# The clinical value of hsa-miR-190b-5p in peripheral blood of pediatric β-thalassemia and its regulation on BCL11A expression

Meihuan Chen[1]☯, Xinrui Wang[1,2]☯, Haiwei Wang🄳[1]☯, Min Zhang[1], Lingji Chen[1], Hong Chen[3], Yali Pan[4], Yanhong Zhang[5], Liangpu Xu[1]*, Hailong Huang🄳[1]*

1 Medical Genetic Diagnosis and Therapy Center of Fujian Maternity and Child Health Hospital College of Clinical Medicine for Obstetrics & Gynecology and Pediatrics, Fujian Medical University, Fujian Provincial Key Laboratory of Prenatal Diagnosis and Birth Defect, Fuzhou, Fujian Province, China, 2 Medical Research Center, Fujian Maternity and Child Health Hospital College of Clinical Medicine for Obstetrics & Gynecology and Pediatrics, Fujian Medical University, Fuzhou, Fujian, China, 3 School of Pharmacy, Fujian Medical University, Fuzhou, Fujian, China, 4 Medical Technology and Engineering College of Fujian Medical University, Fuzhou, Fujian Province, China, 5 Fujian University of Traditional Chinese Medicine, Fuzhou, Fujian Province, China

☯ These authors contributed equally to this work.
* xiliangpu@fjmu.edu.cn (LX); huanghailong@fjmu.edu.cn (HH)

**Data Availability Statement:** All relevant data are within the manuscript and its Supporting Information files.

## Abstract

### Background

The B cell CLL/lymphoma 11A (BCL11A) is a key regulator of hemoglobin switching in β-thalassemia (β-thal). Previous study has suggested that dysregulated microRNAs are involved in the regulation of BCL11A expression. The aim of this study was to investigate the clinical value of hsa-miR-190b-5p in β-thal, and to confirm the regulatory effect of hsa-miR-190b-5p on BCL11A expression.

### Methods

The peripheral blood of 25 pediatric β-thal patients and 25 healthy controls were selected, and qRT-PCR was used to analyze the levels of hsa-miR-190b-5p and BCL11A mRNA. The relationship between hsa-miR-190b-5p expression and hematological parameters was assessed by Pearson's correlation test. The diagnostic power of hsa-miR-190b-5p was evaluated by ROC curves analysis. The direct integration between hsa-miR-190b-5p and BCL11A 3'-UTR was confirmed by luciferase reporter assay.

### Results

Hsa-miR-190b-5p expression in pediatric β-thal was upregulated, and negatively correlated with the MCH and HbA levels, but positively correlated with the HbF level. Hsa-miR-190b-5p showed a good diagnostic capability for pediatric β-thal equivalent to that of HbA$_2$ (AUC: 0.760 vs. 0.758). Moreover, the levels of BCL11A mRNA in pediatric β-thal were decreased, and hsa-miR-190b-5p had a negative correlation with BCL11A mRNA expression ($r$ = -0.403). BCL11A was a target gene of hsa-miR-190b-5p. The mRNA and protein levels of BCL11A were diminished by introduction of hsa-miR-190b-5p, whereas its expression was upregulated by knockdown of hsa-miR-190b-5p.

**Funding:** This study was financially supported by National Natural Science Foundation of China Ideas Grant (81970170) and Joint Funds for the innovation of science and Technology, Fujian province Ideas Grant (2021Y9174) awarded to HLH. This study was also financially supported by Joint Funds for the innovation of science and Technology, Fujian province Ideas Grant (2021Y9173) awarded to MHC. HHL designed the study and supervised the work. MHC collected the blood samples, performed the data analysis and wrote the manuscript.

**Competing interests:** The authors declare that they have no conflicts of interest.

## Conclusions

Hsa-miR-190b-5p expression was upregulated in pediatric β-thal and might be an effective diagnostic biomarker. BCL11A was negatively regulated by hsa-miR-190b-5p, which might provide new target for the treatment of pediatric β-thal.

## Introduction

β-thalassemia (β-thal) is an autosomal recessive hereditary anemia, characterized by the disruption of β-globin expression. β-thal is highly prevalent in Mediterranean countries, including the Middle East, Central Asia, India and Southern China [1, 2]. The phenotypes of β-thal are variable ranging from severe anemia to clinically asymptomatic individuals because of the wide spectrum of mutations in a homozygous or compound heterozygous state [3]. The treatments of β-thal include regular transfusions and hematopoietic stem cell allogenic transplant. However, there are many challenges and limitations in the currently available conventional therapies [4]. In the past few years, epigenetic manipulations and genomic editing have been appeared as novel therapeutic approaches for β-thal [5–8].

The decreasing of β-globin in β-thal compensatory reactivates γ-globin expression and fetal hemoglobin (HbF) synthesis. The transcription factor B-cell lymphoma/leukemia 11A (BCL11A) is a key regulator of hemoglobin switching and HbF silencer in adult. BCL11A directly binds to the promoter regions of HbF and inhibits its expression [9, 10]. In β-thal patients, the expression levels of BCL11A are decreased to reactivate HbF synthesis, and targeting BCL11A represents an important therapeutic strategy [11]. In recently, several other transcription factors show regulatory effect on HbF expression through BCL11A, such as krueppel-like factor 1 (KLF1), activation of transcription factor 4 (ATF4), and MYB proto-oncogene (MYB). Therefore, an increased study about the regulation of BCL11A may lead to find alternative treatments in β-thal.

MicroRNAs (miRNAs) are a category of conserved, small non-coding RNA molecules (21~25 nucleotides in length) [12, 13]. Increasing studies have suggested that dysregulated miRNAs play potential regulatory roles in β-thal by interaction with BCL11A. For instance, increased hsa-miR-30a expression is associated with decreased BCL11A expression and elevated γ-globin and HbF levels in β-thal and miR-30a regulates γ-globin expression through targeting BCL11A [14]. Another study confirms that BCL11A can be directly targeted by hsa-miR-210 and hsa-miR-486-3p in human erythroid cells [15, 16]. However, the relationship between the vast majority of miRNAs and BLC11A expression is unclear. Previously, we analyzed the abnormal regulated miRNAs in peripheral blood of pediatric β-thal by miRNA sequencing and found hsa-miR-190b-5p expression was decreased [17]. Here, we enrolled more clinical samples to further confirm hsa-miR-190b-5p expression in pediatric β-thal and to investigate its clinical value and its regulation on BCL11A expression. The flow chart recapitulating the present study was shown in Fig 1. Our study is critical to further elucidate the epigenetic regulation of BCL11A in pediatric β-thal.

## Materials and methods

### Study participants

This study was conducted in accordance with the Declaration of Helsinki and approved by the Ethics Review Committee of Fujian Province Maternity and Child Health Hospital (approval

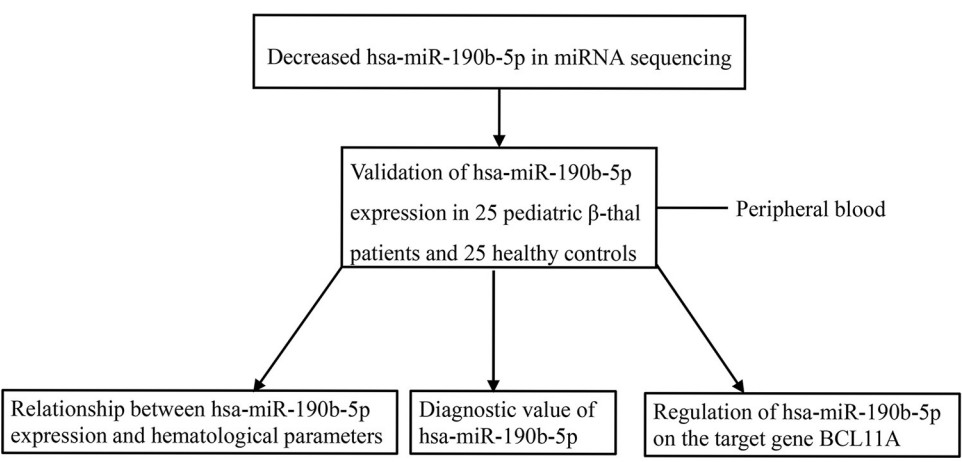

**Fig 1. The flow chart of the study.**

no. 073, 2019). Written informed consent was obtained from all participants' parents following a detailed description of the purpose of the study and the information that could identify individual participants had been accessed during data collection. 25 pediatric β-thal patients(11 pediatric β-thal patients without any transfusions for at least the past month prior to sample collection and 14 pediatric β-thal patients with transfusion history before the past month prior to sample collection) and 25 age paired healthy controls were selected in this study. Inclusion criteria: 1) pediatric patients diagnosed with β-thal by genetic diagnosis; 2) None of the patients were under treatment with hydroxyurea. The genotypes of all subjects were detected using reverse dot blot hybridization (RDB) with β-thal gene detection kits (Shenzhen Yishengtang Biological Products Co., Ltd, Shenzhen, China) following the manufacturer's instructions [18]. The data collected in this study was from January 2020 to December 2021 and the data accessed for research purposes was from September 2021 to November 2022.

## Samples collection

5 ml of peripheral blood was collected from selected subjects and saved with PAXgene Blood RNA Kits (Qiagen, Hilden, Germany) for further RNA isolation. Blood cell parameters were analyzed on a Sysmex XN-3000 automatic hematology analyzer (Sysmex, Shanghai, China). The components and levels of hemoglobin were analyzed using an automated capillary electrophoresis system (version 6.2; Sebia, Paris, France).

## RNA extraction

Total RNA from peripheral blood was isolated using a miRNeasy Mini Kit (Qiagen, Hilden, Germany) according to the manufacturer's protocol. The concentration and purity of the RNA samples were assessed as 260/280 nm and 260/230 nm ratios, respectively and were measured by a Bio Photometer MULTISKAN GOc.

## Quantitative real-time PCR (qRT-PCR)

The cDNA of miRNAs (hsa-miR-190b-5p and U6) was synthesized with a Mir-X™ First Strand Synthesis kit (Takara Bio, Inc., Japan) according to the manufacturer's instructions. The cDNA of mRNAs (BCL11A and β-actin) was synthesized with a PrimeScript™ RT-PCR kit (Takara Bio, Inc., Japan) according to the manufacturer's instructions. qPCR was performed

with the cDNA on a StepOne Real-time PCR System (Applied Biosystems, Foster City, CA, USA). The following primers were used for qPCR: BCL11A sense: 5′-ACAGGAACACATAG CAGATAAAC-3′ and antisense: 5′-TATTCTGCACTCATCCCAGG-3′; β-actin sense: 5′-GCACAGAGCCTCGCCTT-3′ and antisense: 5′-GTTGTCGACGACGAGCG-3′. β-actin was used as the internal control for mRNA quantification and U6 was used as the internal control for miRNA quantification. Relative expression levels of hsa-miR-190b-5p and BCL11A were calculated using the comparative cycle threshold method.

## Cell culture and cell transfection

Human 293T cells were purchased from Type Culture Collection of Chinese Academy of Sciences (Shanghai, China), and were cultured in Dulbecco's modified Eagle's medium (DMEM; GibcoBRL, Gaithersburg, MD, USA) supplemented with 10% fetal bovine serum (GibcoBRL, Gaithersburg, MD, USA). All cells were maintained at 37°C in a humidified atmosphere containing 5% $CO_2$. The hsa-miR-190b-5p mimics (5′-UGAUAUGUUUGAUAUUGGGUUG-3′) and inhibitors (5′-CAACCCAAUAUCAAACAUAUCA-3′) and its negative control mimics (5′-UUCUCCGAACGUGUCACGUTT-3′) and inhibitors (5′-CAGUACUUUUGUGUAGUA CAA-3′) were purchased from Genepharma (Genepharma Inc., Shanghai, China). Cell transfection was performed with these oligonucleotides at a concentration of 100 nM using Lipofectamine 3000 reagent (Thermo Scientific, Waltham, MA, USA) according to the manufacturer's protocol. Total RNA was extracted at 48 h post-transfection and used for qRT-PCR analysis.

## Western blots

Total protein from 293T cells was extracted with RIPA lysis buffer (Thermo Scientific, Waltham, MA, USA). Protein concentrations were measured using a BCA protein assay kit (Beyotime Biotechnology, Shanghai, China). The same amount of protein samples (35 μg) was loaded onto an SDS-polyacrylamide electrophoresis gels and then transferred onto polyvinylidene difluoride membranes (Merck, Billerica, MA, USA). After blocked in 5% skimmed milk for 2 h, the membranes were incubated with primary BCL11A antibody (Abcam, Cambridge, MA, USA; ab19489; 1:1000) overnight at 4°C and with HRP-conjugated rabbit anti-mouse secondary antibody for 60 min at room temperature. The β-actin antibody (Abclonal, Wuhan, China; AC004; 1:1000) was used as internal control. The blots were detected using ECL chemiluminescent reagent (Beyotime Biotechnology, Shanghai, China).

## Plasmids construction and luciferase reporter assay

Three online database for miRNA target prediction, including miRanda (http://www.microrna.org), TargetScan (http://www.targetscan.org) and miRDB (http://mirdb.org) were used for predicting the binding sites between hsa-miR-190b-5p and human BCL11A mRNA 3'-UTR. The wild-type (WT) human BCL11A 3'-UTR fragments (499–506 loci: ACATATC; 2553–2559 loci: ACATATC) including hsa-miR-190b-5p-binding sites were synthesized by Sangon Biotech (Shanghai, China). Also, the mutant (MUT) human BCL11A mRNA 3'-UTR fragments (ACATATC>TGTATAG) were synthesized. These fragments were inserted into the multiple cloning sites of pmiR-RB-REPORT™, yielding WT1 and WT2, MUT1 and MUT2 luciferase reporter vectors. For the luciferase reporter assay, 293T cells were co-transfected with these reporter vectors and hsa-miR-190b-5p mimics. After 48 h, cells were collected and the luciferase activity was measured by using Dual-Luciferase Reporter Assay System (Promega, Madison, WI, USA) according to the kit instructions.

## Statistical analysis

The GraphPad Prism 9 software (GraphPad Software Inc., San Diego, CA, USA) and SPSS 25.0 software (SPSS, Chicago, IL, US, USA) were used for statistical analyses. Measurement data were expressed as the mean ± standard deviation (SD). The normality of the sample distributions was calculated by the Kolmogorov-Smirnoff test. ANOVA and bonferroni post hoc test were used to evaluate the differences in variables between pediatric β-thal patients with or without transfusion and healthy controls. The correlations between the expression levels of hsa-miR-190b-5p and hematological parameters, BCL11A mRNA expression were evaluated by Pearson's correlation test. The receiver operating characteristic (ROC) curves analysis and the area under curve (AUC) were carried on to detect hsa-miR-190b-5p as a biomarker for differentiating pediatric β-thal from healthy controls. A value of $p < 0.05$ was considered statistically significant.

## Results

### Comparison of hematological parameters of pediatric β-thal patients and healthy controls

No statistical difference was found in age (average age: 7.14 ± 2.28 and 7.27 ± 2.00 vs. 7.24 ± 2.02) and sex distribution (male/female: 8/6 and 7/4 vs. 14/11) between pediatric β-thal patients with or without transfusion and healthy controls. The data of hematological parameters were presented in Table 1. Compared with healthy controls, the levels of red blood cells (RBC) and hemoglobin (Hb) were significantly lower in pediatric β-thal patients ($p < 0.05$). Moreover, the level of mean corpuscular volume (MCV), mean corpuscular hemoglobin (MCH) and hemoglobin A (HbA) were significantly lower in pediatric β-thal patients without transfusion ($p < 0.05$). On the contrary, the levels of HbF in pediatric β-thal patients without transfusion were significantly higher than healthy controls ($p < 0.05$). However, there was no statistical difference of MCV, MCH, HbA, $HbA_2$ and HbF between pediatric β-thal patients with transfusion and healthy controls.

### Upregulated hsa-miR-190b-5p expression in peripheral blood from pediatric β-thal patients

Previously, by miRNA sequencing, the expression levels of hsa-miR-190b-5p was found to be downregulated in peripheral blood from 5 cases of pediatric β-thal patients. Here, we aimed to

**Table 1. The hematological parameters of healthy controls and pediatric β-thal patients.**

| Parameters | Healthy controls (n = 25) | Pediatric β-thal patients with transfusion (n = 14) | Pediatric β-thal patients without transfusion (n = 11) | F value | p value |
|---|---|---|---|---|---|
| RBC (×10$^{12}$/L) | 4.63 ± 0.22 | 3.53 ± 0.51[a*] | 3.26 ± 0.68 [b*] | 48.315 | <0.001 |
| Hb (g/L) | 133.12 ± 5.73 | 92.91 ± 20.33 [a*] | 86.55 ± 14.36 [b*] | 70.128 | <0.001 |
| MCV (fL) | 83.63 ± 2.20 | 86.77 ± 3.39 | 68.85 ± 20.59 [b*] | 10.648 | <0.001 |
| MCH (pg) | 28.78 ± 0.83 | 28.46 ± 0.98 | 24.51 ± 1.61 [b*] | 62.253 | <0.001 |
| HbA (%) | 96.94 ± 0.59 | 94.22 ± 1.53 | 81.74 ± 12.26 [b*] | 23.419 | <0.001 |
| $HbA_2$ (%) | 2.80 ± 0.17 | 3.04 ± 0.19 | 4.09 ± 3.27 | 2.390 | 0.105 |
| HbF (%) | 0.17 ± 0.35 | 2.74 ± 1.51 | 14.17 ± 12.77 [b*] | 18.384 | <0.001 |

β-thal: β-thalassemia, RBC: red blood cells, Hb: hemoglobin, MCV: mean corpuscular volume, MCH: mean corpuscular hemoglobin, HbA: hemoglobin A, $HbA_2$: hemoglobin $A_2$, HbF: fetal hemoglobin. Data expressed as mean ± standard deviation (SD).

a: pediatric β-thal patients with transfusion compared with healthy controls

b: pediatric β-thal patients without transfusion compared with healthy controls

*$p < 0.05$.

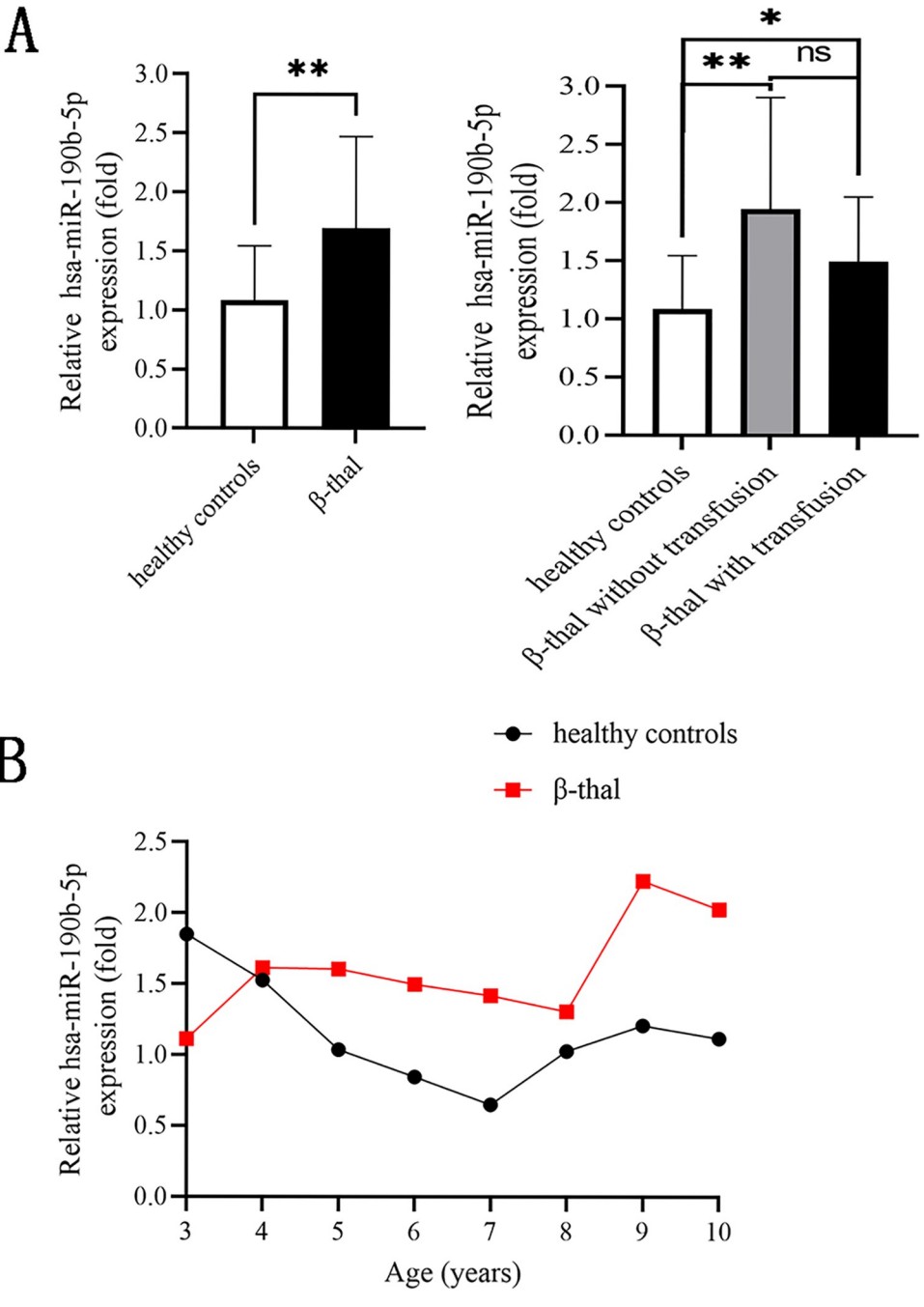

**Fig 2. hsa-miR-190b-5p expression in peripheral blood from pediatric β-thal patients.** (A) qRT-PCR analysis showed that hsa-miR-190b-5p expression levels were upregulated in pediatric β-thal patients in comparison with healthy controls. (B) The dynamic of hsa-miR-190b-5p expression levels at different ages of healthy controls and pediatric β-thal patients. miR: microRNA, β-thal: β-thalassemia, qRT-PCR: quantitative real-time PCR. Data expressed as mean ± standard deviation (SD). Compared with healthy controls, $^{*}p<0.05$,$^{**}p<0.01$ and $^{ns}p>0.05$.

confirmed hsa-miR-190b-5p expression in peripheral blood from 25 pediatric β-thal patients. Surprisingly, we found hsa-miR-190b-5p expression in pediatric β-thal patients with or without transfusion were higher than that in healthy controls ($p<0.05$, Fig 2A). Moreover, the expression levels of hsa-miR-190b-5p in pediatric β-thal patients with transfusion were a bit

lower, but not significantly, than that in pediatric β-thal patients without transfusion ($p>0.05$, Fig 2A). The dynamic of hsa-miR-190b-5p expression in pediatric β-thal patients with different ages was shown in Fig 2B, confirming upregulated hsa-miR-190b-5p levels in pediatric β-thal patients.

### Relationship between hsa-miR-190b-5p expression and hematological parameters

To further analyze the roles of hsa-miR-190b-5p in the development of β-thal, the relationship between hsa-miR-190b-5p expression and hematological parameters in pediatric β-thal patients without transfusion were determined. Our results showed that the expression level of hsa-miR-190b-5p was negatively correlated with the MCH ($r = -0.720$) and HbA ($r = -0.732$) levels, but was positively correlated with the HbF level ($r = 0.740$) in pediatric β-thal patients without transfusion ($p<0.05$, Table 2). The above data implied that upregulated hsa-miR-190b-5p expression might involve in the pathogenesis of pediatric β-thal, specifically in the regulation of HbF.

### Evaluation of HbA2 and hsa-miR-190b-5p for the diagnosis of pediatric β-thal

Subsequently, we performed ROC curves analysis to examine the potential value of $HbA_2$ in diagnosis of pediatric β-thal. The AUC value was 0.758 (95% CI: 0.585–0.931), hinting $HbA_2$ as an effective biomarker for pediatric β-thal ($p<0.01$, Fig 3A). Also, hsa-miR-190b-5p showed similar diagnostic capability to $HbA_2$ in discriminating pediatric β-thal patients from healthy controls, with AUC of 0.760 (95% CI: 0.623–0.897) ($p<0.01$, Fig 3B). Moreover, when hsa-miR-190b-5p combined with $HbA_2$, the AUC value increased to 0.882 (95% CI: 0.771–0.994), which was significantly higher than hsa-miR-190b-5p and $HbA_2$ alone ($p<0.001$, Fig 3C). Thus, hsa-miR-190b-5p had a promising potential as a biomarker for pediatric β-thal.

### Hsa-miR-190b-5p directly interacted with the 3′-UTR of BCL11A mRNA

To identify the correlation between hsa-miR-190b-5p and BCL11A, the expression of BCL11A mRNA in 25 pediatric β-thal patients were detected. As expected, the expression levels of BCL11A mRNA were decreased in pediatric β-thal patients compared to healthy controls ($p<0.05$, Fig 4A and 4B). Additionally, the expression of hsa-miR-190b-5p had a negative correlation with BCL11A mRNA expression in pediatric β-thal patients ($r = -0.403$, $p<0.05$, Fig 4C).

In order to confirm the binding sites (499–506 loci: ACATATC; 2553–2559 loci: ACATATC) of hsa-miR-190b-5p on 3′-UTR of BCL11A mRNA, the WT1 and WT2 reporter vectors were constructed (Fig 5A), and luciferase reporter assay was applied in 293T cells by co-transfection of these reporter vectors and hsa-miR-190b-5p mimics. qRT-PCR verified that the expression levels of hsa-miR-190b-5p were significantly increased in 293T cells after

**Table 2. The relationship between hsa-miR-190b-5p expression and hematological parameters in pediatric β-thal patients without transfusion.**

|  | RBC | Hb | MCV | MCH | HbA | HbA2 | HbF |
|---|---|---|---|---|---|---|---|
| Coefficient | 0.134 | -0.188 | 0.087 | -0.720 | -0.732 | -0.148 | 0.740 |
| *p* value | 0. 695 | 0.579 | 0.800 | 0.012* | 0.011* | 0.665 | 0.009** |

*$p<0.05$

**$p<0.01$.

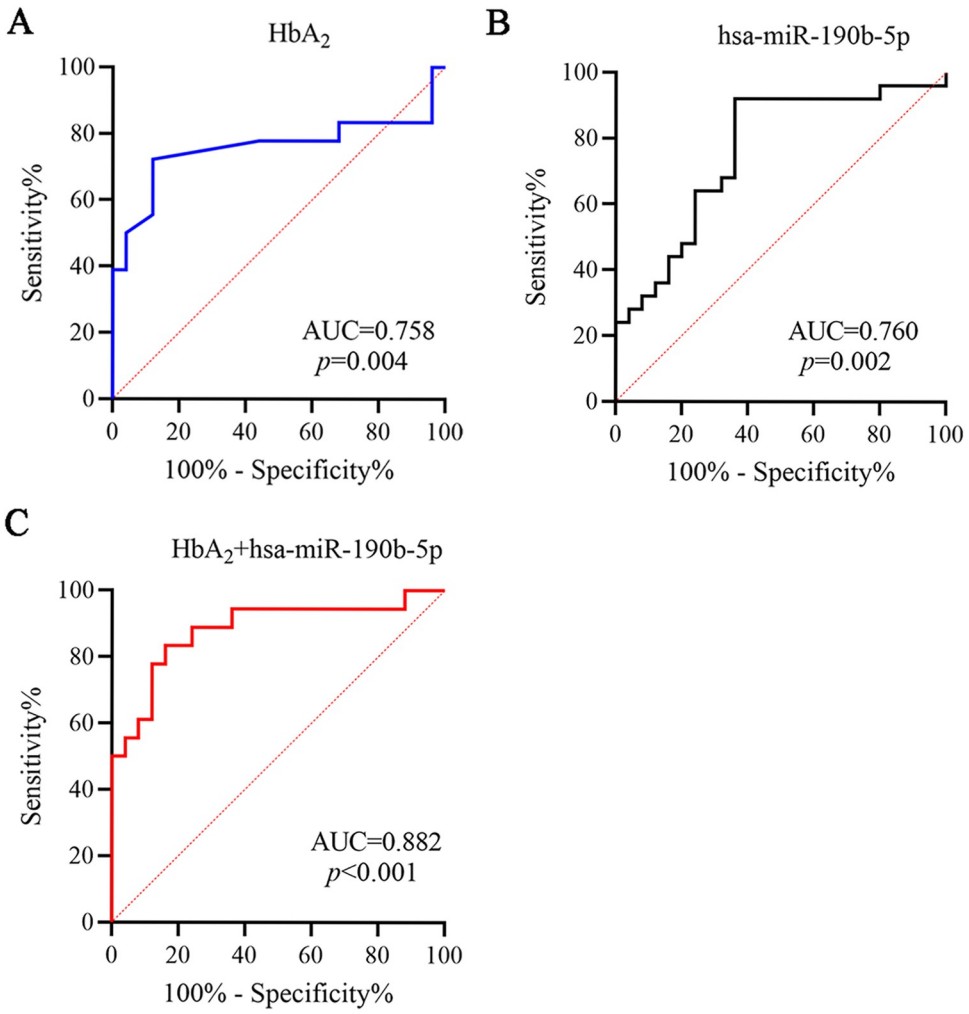

**Fig 3. hsa-miR-190b-5p as a diagnostic biomarker for pediatric β-thal patients.** ROC curves analysis of the AUC values in HbA$_2$ alone (A), hsa-miR-190b-5p alone (B), and combination of both (C) for discriminating pediatric β-thal patients from healthy controls. ROC: receiver operating characteristic, AUC: the area under curve.

transfection of hsa-miR-190b-5p mimics ($p<0.001$, Fig 5B). Compared with the negative control mimics, hsa-miR-190b-5p mimics significantly decreased the relative luciferase activity of WT2 ($p<0.05$, Fig 5C), but no significant difference was found in the relative luciferase activity of WT1 (Fig 5D). Inversely, when the WT1 and WT2 were mutated to MUT1 and MUT2, the relative luciferase activity was unaffected by either hsa-miR-190b-5p mimics or negative control mimics. The data demonstrated that BCL11A was a target gene of hsa-miR-190b-5p, and hsa-miR-190b-5p could directly bind to the 2553–2559 loci of BCL11A mRNA 3'-UTR.

## BCL11A expression was negatively regulated by hsa-miR-190b-5p

Finally, we determined the expression of BCL11A regulated by hsa-miR-190b-5p. Results found that hsa-miR-190b-5p mimics reduced the levels of BCL11A mRNA and protein in 293T cells ($p<0.05$, Fig 6A and 6B). qRT-PCR verified that hsa-miR-190b-5p inhibitors suppressed hsa-miR-190b-5p expression in 293T cells ($p<0.01$, Fig 6C). As shown in Fig 6D and 6E, in contrast to negative control inhibitors, the expression levels of BCL11A mRNA and

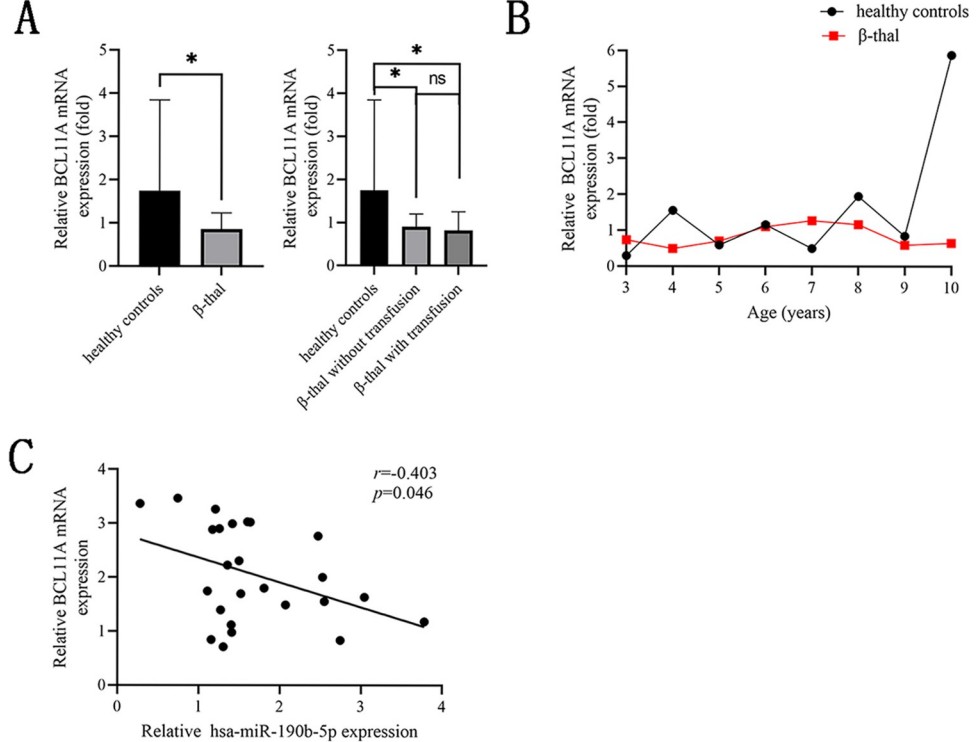

**Fig 4. BCL11A mRNA expression in peripheral blood from pediatric β-thal patients.** (A) qRT-PCR analysis revealed that the expression levels of BCL11A mRNA were decreased in pediatric β-thal patients compared to healthy controls. (B) The dynamic of hsa-miR-190b-5p expression levels at different ages of healthy controls and pediatric β-thal patients. (C) Pearson's correlation test analysis of the correlation between hsa-miR-190b-5p level and BCL11A mRNA expression in pediatric β-thal patients. BCL11A: B-cell lymphoma/leukemia 11A. Data expressed as mean ± SD. Compared with healthy controls, *$p < 0.05$ and $^{ns}p > 0.05$.

protein were significantly increased in 293T cells after transfection of hsa-miR-190b-5p inhibitors ($p < 0.05$). Together, these results suggested that hsa-miR-190b-5p could negatively regulate BCL11A expression.

## Discussion

Dysregulation of miRNAs had been closely related to the regulation role in the development of β-thal and acted as an essential modulatory molecule in γ-globin expression. In this study, we firstly demonstrated that the expression levels of hsa-miR-190b-5p were significantly upregulated in pediatric β-thal patients and hsa-miR-190b-5p expression was associated with the severity of several hematological parameters, including MCH, HbA and HbF. Moreover, hsa-miR-190b-5p showed a similar diagnostic value as HbA$_2$ for the diagnosis of pediatric β-thal. In addition, BCL11A expression was found to be negatively regulated by hsa-miR-190b-5p, and the 2553–2559 loci of BCL11A mRNA 3'-UTR was the binding sites. These data strongly implied that hsa-miR-190b-5p might be a novel biomarker in the diagnosis and therapy of pediatric β-thal.

Hsa-miR-190b was localized on q21.3 of human Chromosome 1, which was abnormally expressed in human diseases, such as multiple cancers, gestational diabetes mellitus, liver ischaemia-reperfusion injury, and non-alcoholic fatty liver disease [19–25]. In previous study, we reported that hsa-miR-190b-5p expression was downregulated in pediatric β-thal patients (mean age at 2.6 ± 1.18 years) by microRNA sequencing [17]. However, the role of hsa-miR-

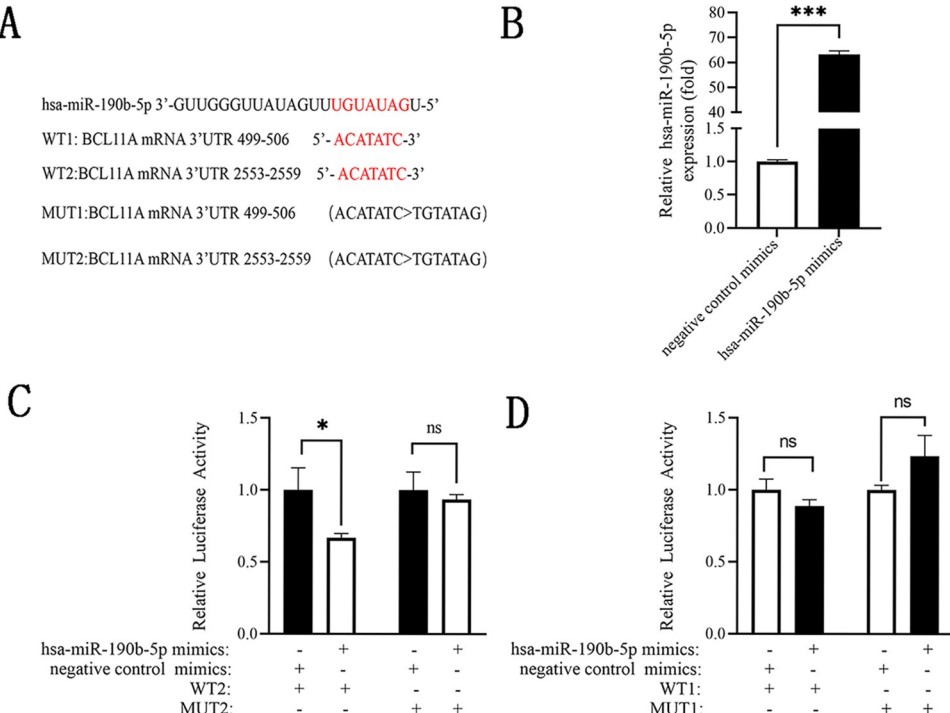

**Fig 5. BCL11A was a target of hsa-miR-190b-5p.** (A) The human BCL11A mRNA 3'-UTR harbored two putative binding sites (499–506 loci: ACATATC; 2553–2559 loci: ACATATC) of hsa-miR-190b-5p. These mutant binding sites (ACATATC>TGTATAG) were shown below. (B) qRT-PCR analysis of hsa-miR-190b-5p expression in 293T cells after transfected with hsa-miR-190b-5p mimics and negative control mimics. (C) Luciferase reporter assay analysis of the relative luciferase activity in 293T cells co-transfected with the WT2 or MUT2 reporter vector and hsa-miR-190b-5p mimics or negative control mimics. (D) hsa-miR-190b-5p mimics had no significant effect on the relative luciferase activity of WT1 and MUT1 reporter vectors. WT: wild type, MUT: mutant. Data expressed as mean ± SD. Compared with negative control mimics, $*p<0.05$, $***p<0.001$, and $^{ns}p>0.05$.

190b in pediatric β-thal remained unknown. In the present study, increased expression levels of hsa-miR-190b-5p were observed in pediatric β-thal patients (mean age at 7.20 ± 2.12 years) in comparison with healthy controls, which was contrary to the previous study [17]. The reason for the contradictory results might be due to the variant expression of hsa-miR-190b-5p in pediatric β-thal patients of different ages. Further, we analyzed the dynamic of hsa-miR-190b-5p expression in pediatric β-thal patients with different ages and found 4 years of age was the cut-off for hsa-miR-190b-5p expression. The hsa-miR-190b-5p expression was lower in pediatric β-thal patients younger than 4 years than healthy controls, which confirmed the low expression of hsa-miR-190b-5p in the previous sequencing. These data demonstrated that hsa-miR-190b was an age-related miRNA in pediatric β-thal. Moreover, when pediatric β-thal patients underwent transfusion, the expression levels of hsa-miR-190b-5p were not statistically different from that without transfusion, inferring the hsa-miR-190b expression might not be affected by transfusion.

HbA$_2$, as a practical biomarker, had been shown a considerable sensitivity in diagnosis of patients with microcytic or hypochromic anemia [26]. However, its specificity for β-thal was not quite ideal. Today, the majority of dysregulated miRNAs serve as diagnostic biomarkers for various of human diseases, such as tumors, hematological disease, cardiovascular disease, and immune system disease [27–30]. In patients with Sarcoma, hsa-miR-190b was identified as a novel prognostic biomarker [31]. In patients with endometrial cancer, hsa-miR-190b showed a high diagnostic capability in differentiating grade 3 of tumor from grade 2 of tumor

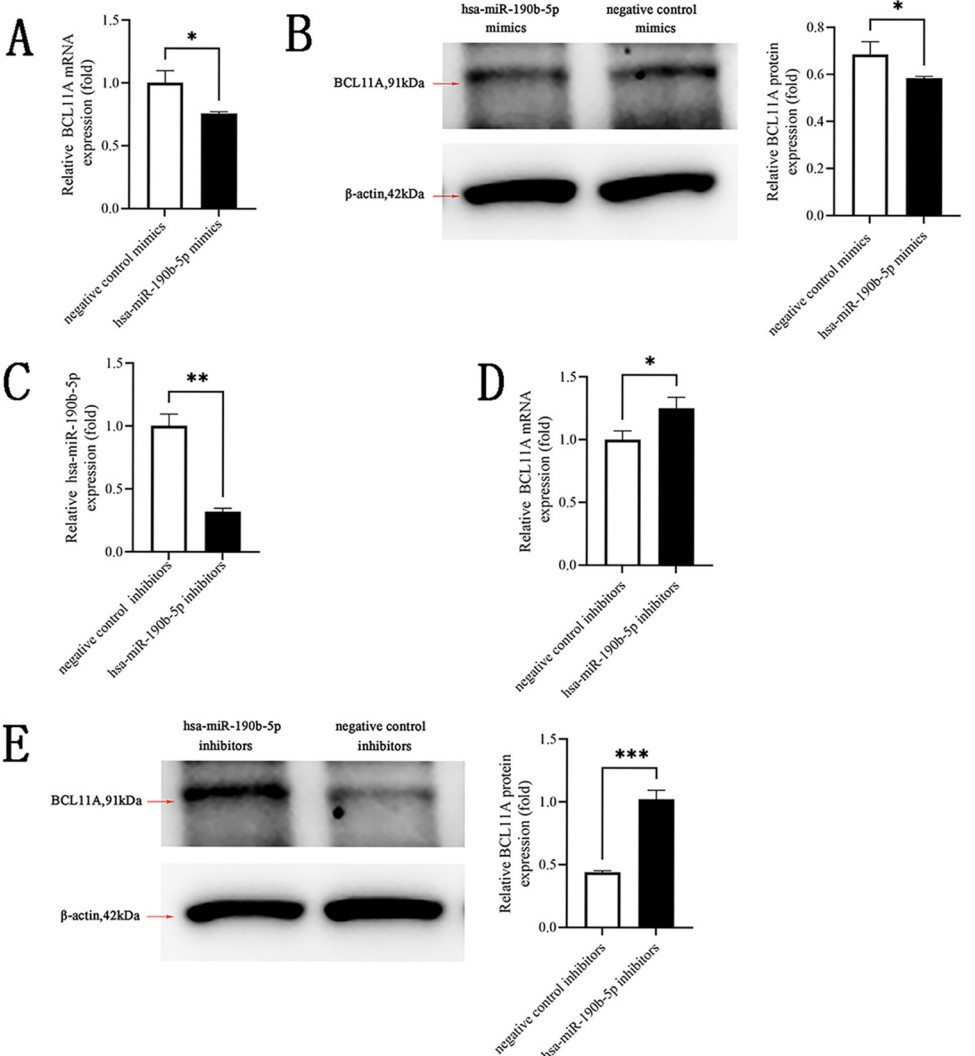

**Fig 6. BCL11A expression was negatively regulated by hsa-miR-190b-5p.** (A, B) qRT-PCR and Western blot analysis of the mRNA and protein expression levels of BCL11A in 293T cells after transfected hsa-miR-190b-5p mimics and negative control mimics. β-actin gene was used as the internal control. (C) hsa-miR-190b-5p expression was decreased in 293T cells by hsa-miR-190b-5p inhibitors. (D,E) qRT-PCR and Western blot analysis of the mRNA and protein expression levels of BCL11A in 293T cells after transfected with hsa-miR-190b-5p inhibitors and negative control inhibitors. Data expressed as mean ± SD. Compared with negative control mimics or negative control inhibitors, $*p<0.05$, $**p<0.01$, $***p<0.001$.

[32]. Additional, Dai et al, demonstrated that hsa-miR-190b served as potential diagnostic and prognostic biomarkers for breast cancer [19]. These reports prompted us to investigate the diagnostic role of hsa-miR-190b in pediatric β-thal. ROC curves analysis showed that the AUC value of hsa-miR-190b was 0.760, which was almost identical AUC value for HbA$_2$ (0.758) in discriminating pediatric β-thal patients from healthy controls. Moreover, hsa-miR-190b-5p combined with HbA$_2$ could significantly increase the value of AUC to 0.882. The above data hinted that decreased hsa-miR-190b-5p could be used as a potential diagnostic biomarker for pediatric β-thal.

When the correlation between hsa-miR-190b-5p expression and hematological parameters in pediatric β-thal patients were evaluated, we found that hsa-miR-190b-5p had a negative

correlation with MCH and HbA levels, but had a positive correlation with the HbF level. Increasing studies have suggested that targeting BCL11A to reactivate HbF expression represented an important therapeutic strategy for β-thal [11]. The BCL11A expression in β-thal had been shown to be regulated by several miRNAs, such as hsa-let-7, hsa-miR-138-5p and hsa-miR-92a-3p [33–35]. Based on the above evidence, we presumed that has-miR-190b-5p might be involved in the regulation of BCL11A. Here, decreased BCL11A mRNA expression was observed in pediatric β-thal patients, which was consistent with previous study by Gholampour MA et al. [14]. Pearson's correlation test found that hsa-miR-190b-5p level was negative correlated with BCL11A levels in pediatric β-thal patients, inferring that BCL11A might be a target of hsa-miR-190b-5p.

Bioinformatics analysis revealed BCL11A mRNA 3'-UTR had two predicting binding sites of hsa-miR-190b-5p, one was located in 499–506 loci and the other was presented in 2553–2559 loci. By sequence comparison, both loci were very conserved in mammals. In order to verify the direct regulatory effect of hsa-miR-190b-5p on BCL11A, the reporter vectors were constructed based on these loci and luciferase reporter assay was performed. Result found that hsa-miR-190b-5p could directly bind to the 2553–2559 loci of BCL11A mRNA 3'-UTR, but not 499–506 loci. More importantly, loss-and gain-of-function approaches substantiated that hsa-miR-190b-5p could negatively regulate BCL11A expression. Therefore, it could be confirmed that BCL11A was a target of hsa-miR-190b-5p in pediatric β-thal.

The shortcomings of the current study were as follows: 1) The sample size enrolled in this study was small, which might have limited generalizability. Large-scale clinical trials were needed to further validate the expression level and clinical value of hsa-miR-190b-5p in pediatric β-thal. 2) The role of hsa-miR-190b-5p in the differentiation of erythroid precursor cells was unknown, which would affect the determination of whether it could be provided as an effective strategy for pediatric β-thal treatment. Further studies were necessary to fucus on its role and relationship with BCL11A in erythroid precursor cells and in extensive animal experiments.

## Conclusions

Our data demonstrated hsa-miR-190b-5p expression was upregulated in pediatric β-thal patients and its expression was correlated with the MCH, HbA, and HbF levels. Hsa-miR-190b-5p might be an effective biomarker for the diagnosis of pediatric β-thal and hsa-miR-190b-5p combined with $HbA_2$ could significantly increase the diagnostic value. BCL11A was a direct target of hsa-miR-190b-5p in pediatric β-thal, indicating hsa-miR-190b-5p/BCL11A pathway might provide new targets for the treatment of pediatric β-thal.

## Supporting information

**S1 Fig. Western blot analysis of the protein expression levels of BCL11A in 293T cells after transfected hsa-miR-190b-5p mimics and inhibitors.**
(TIF)

**S1 Raw images.**
(PDF)

## Acknowledgments

The authors would like to thank Dr Xinhua Zhang from Department of Hematology, 923 rd Hospital of the People's Liberation Army, Nanning, Guangxi 530021, China, for directing the blood samples collection.

## Author Contributions

**Conceptualization:** Min Zhang, Liangpu Xu.

**Data curation:** Lingji Chen, Hong Chen, Yali Pan, Yanhong Zhang.

**Writing – original draft:** Meihuan Chen.

**Writing – review & editing:** Xinrui Wang, Haiwei Wang, Hailong Huang.

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
