## [Decision Letter · Decision Letter 0]

13 Jun 2023

PONE-D-23-10873The clinical value of hsa-miR-190b-5p in peripheral blood of pediatric β-thalassemia and its regulation on BCL11A expressionPLOS ONE

Dear Dr. Huang,

Thank you for submitting your manuscript to PLOS ONE. After careful consideration, we feel that it has merit but does not fully meet PLOS ONE’s publication criteria as it currently stands. Therefore, we invite you to submit a revised version of the manuscript that addresses the points raised during the review process.

We look forward to receiving your revised manuscript.

Kind regards,

J Francis Borgio, Ph.D.,

Academic Editor

PLOS ONE

Journal Requirements:

In your cover letter, please note whether your blot/gel image data are in Supporting Information or posted at a public data repository, provide the repository URL if relevant, and provide specific details as to which raw blot/gel images, if any, are not available. Email us at plosone@plos.org if you have any questions

4.  PLOS requires an ORCID iD for the corresponding author in Editorial Manager on papers submitted after December 6th, 2016. Please ensure that you have an ORCID iD and that it is validated in Editorial Manager. To do this, go to ‘Update my Information’ (in the upper left-hand corner of the main menu), and click on the Fetch/Validate link next to the ORCID field. This will take you to the ORCID site and allow you to create a new iD or authenticate a pre-existing iD in Editorial Manager. Please see the following video for instructions on linking an ORCID iD to your Editorial Manager account: https://www.youtube.com/watch?v=_xcclfuvtxQ.

Additional Editor Comments :

Authors should list the limitations of the study properly in the discussion. When listing the limitations of a study, authors should be specific and provide details about how the limitations might have affected the findings. They should also discuss how the limitations could be addressed in future research.

Reviewers' comments:

Reviewer's Responses to Questions

**Comments to the Author**

1. Is the manuscript technically sound, and do the data support the conclusions?

Reviewer #1: Yes

Reviewer #2: Yes

2. Has the statistical analysis been performed appropriately and rigorously? 

Reviewer #1: I Don't Know

Reviewer #2: Yes

3. Have the authors made all data underlying the findings in their manuscript fully available?

Reviewer #1: Yes

Reviewer #2: Yes

4. Is the manuscript presented in an intelligible fashion and written in standard English?

Reviewer #1: Yes

Reviewer #2: Yes

5. Review Comments to the Author

Reviewer #1: The manuscript has been thoroughly reviewed, and no further comments or revisions have been proposed, indicating a recommendation for acceptance. However, it is important to address two specific issues raised during the review process.:

Firstly, the study's small sample size should be acknowledged as a potential limitation and discussed in the appropriate section.

Additionally, the quality of the figures needs improvement, and it is advised to replace them with higher-quality images to enhance visual clarity and data presentation

Abstract: clear and concise abstract that summarizes the main points of the manuscript.

The introduction provides a clear and concise overview of the background, significance, and objectives of the study

Methods: The methods and experimental procedures described in sufficient detail to allow reproducibility. Clearly state the study design, sample size, data collection methods, and statistical analyses used.

The results Tables, figures, and graphs used in a logical and organized manner, when appropriate. Relevant statistical analyses included and the key findings highlighted.

Discussion: The results interpreted and discussed properly.

Reviewer #2: This manuscript satisfies the PLOS ONE criteria for publication as it is an original manuscript with added knowledge in the field. The results presented have not been published elsewhere. The authors well described the methodology and results with sufficient details. This study has received ethical clearance from the Ethics Review Committee of Fujian Province Maternity and Child Health Hospital (approval no. 073, 2019), and this study has been conducted with adherence to the Declaration of Helsinki. This manuscript has provided new information on β-thal disease by highlighting the role of the hsa-miR-19b-5p in ameliorating the β-thal through its direct action on BCL11A to downregulate BCL11A and thus increase the HbF level in β-thal patients. The potential of hsa-miR-190b-5p as a biomarker for β-thal was evaluated by the authors, who suggested that hsa-miR-190b-5p is similarly important as HbA2 for discriminating pediatric β-thal patients from healthy controls. The authors also proved the capability of hsa-miR-19b-5p direct binding to BCL11A. The authors have clearly and thoroughly discussed their findings with relevant references. Some grammatical errors may be corrected prior to publication. 1) Page 8, recheck this sentence: "Compared with healthy controls, the levels of red blood cells (RBC) and hemoglobin (Hb) in pediatric β-thal patients (p<0.001)."; 2) Page 12, recheck this sentence: "The above data hinted that that decreased hsa-miR-190b-5p has potential as a diagnostic biomarker for pediatric β-thal."; 3) Figure 1, recheck this sentence: "Regulation of hsa-miR-190b-5p on the garget gene BCL11A.".

6. PLOS authors have the option to publish the peer review history of their article (what does this mean?). If published, this will include your full peer review and any attached files.

Reviewer #1: No

Reviewer #2: No

---

## [Author Response · Author response to Decision Letter 0]

13 Jul 2023

Response letter

[Jun 29 2023]

Dear Editor and reviewers,

First of all, thank you very much for giving us an opportunity to revise our manuscript. We appreciate editor and reviewers for the comments concerning our manuscript entitled “The clinical value of hsa-miR-190b-5p in peripheral blood of pediatric β-thalassemia and its regulation on BCL11A expression” (ID: PONE-D-23-10873). Those comments are all valuable and very helpful for revising and improving our paper. We have studied comments carefully and have made correction which we hope meet with approval. We have thoroughly revised the manuscript accordingly, and changes corresponding to the issues were highlighted in Red in the revised version.

 We very much appreciated the valuable comments and suggestions made by the reviewers. We hope that the changes have adequately addressed their concerns. Thank you very much for the time and effort you have spent to help us improve this manuscript.

Sincerely yours,

Hailong Huang

Medical Genetic Diagnosis and Therapy Center of Fujian Maternity and Child Health Hospital College of Clinical Medicine for Obstetrics & Gynecology and Pediatrics, Fujian Medical University, Fujian Provincial Key Laboratory of Prenatal Diagnosis and Birth Defect, Fuzhou 350001, Fujian Province, China

Email: huanghailong@fjmu.edu.cn

Responses to the Editor

Response: We ensure that our manuscript meets PLOS ONE's style requirements, including those for file naming.

Response: We ensure that the ethics statement only appears in the Methods section of our manuscript.

3. PLOS ONE now requires that authors provide the original uncropped and unadjusted images underlying all blot or gel results reported in a submission’s figures or Supporting Information files. This policy and the journal’s other requirements for blot/gel reporting and figure preparation are described in detail at https://journals.plos.org/plosone/s/figures#loc-blot-and-gel-reporting-requirements and https://journals.plos.org/plosone/s/figures#loc-preparing-figures-from-image-files. When you submit your revised manuscript, please ensure that your figures adhere fully to these guidelines and provide the original underlying images for all blot or gel data reported in your submission. See the following link for instructions on providing the original image data: https://journals.plos.org/plosone/s/figures#loc-original-images-for-blots-and-gels. In your cover letter, please note whether your blot/gel image data are in Supporting Information or posted at a public data repository, provide the repository URL if relevant, and provide specific details as to which raw blot/gel images, if any, are not available. Email us at plosone@plos.org if you have any questions

Response: We ensure that our original uncropped and unadjusted images underlying blot/gel image data are shown in Supporting Information.

4. PLOS requires an ORCID iD for the corresponding author in Editorial Manager on papers submitted after December 6th, 2016. Please ensure that you have an ORCID iD and that it is validated in Editorial Manager. To do this, go to ‘Update my Information’ (in the upper left-hand corner of the main menu), and click on the Fetch/Validate link next to the ORCID field. This will take you to the ORCID site and allow you to create a new iD or authenticate a pre-existing iD in Editorial Manager. Please see the following video for instructions on linking an ORCID iD to your Ed Response: We ensure that the corresponding author ORCID Id (0000-0001-5775-5082) is validated in Editorial Manager account.

Additional Editor Comments :

5.Authors should list the limitations of the study properly in the discussion. When listing the limitations of a study, authors should be specific and provide details about how the limitations might have affected the findings. They should also discuss how the limitations could be addressed in future research.

Response: We are very appreciated for this comment. We have listed the limitations of the study in the discussion in Page 16, line 10-17 as “The shortcomings of the current study were as follows: 1) The sample size enrolled in this study was small, which might have limited generalizability. Large-scale clinical trials were needed to further validate the expression level and clinical value of hsa-miR-190b-5p in pediatric β-thal. 2) The role of hsa-miR-190b-5p in the differentiation of erythroid precursor cells was unknown, which would affect the determination of whether it could be provided as an effective strategy for pediatric β-thal treatment. Further studies were necessary to fucus on its role and relationship with BCL11A in erythroid precursor cells and in extensive animal experiments.”

Responses to Reviewer #1’s comments

Reviewer #1: The manuscript has been thoroughly reviewed, and no further comments or revisions have been proposed, indicating a recommendation for acceptance. However, it is important to address two specific issues raised during the review process.:

Firstly, the study's small sample size should be acknowledged as a potential limitation and discussed in the appropriate section.

Response: We are very appreciated for this comment. We have listed the “the study's small sample size” as a limitation of the study in the discussion in Page 16, line 10-13 as “The shortcomings of the current study were as follows: 1) The sample size enrolled in this study was small, which might have limited generalizability. Large-scale clinical trials were needed to further validate the expression level and clinical value of hsa-miR-190b-5p in pediatric β-thal.”

Additionally, the quality of the figures needs improvement, and it is advised to replace them with higher-quality images to enhance visual clarity and data presentation

Response: We are very appreciated for this comment. We have replaced these figures with higher-quality to enhance visual clarity and data presentation.

Responses to Reviewer #2’s comments

Reviewer #2: This manuscript satisfies the PLOS ONE criteria for publication as it is an original manuscript with added knowledge in the field. The results presented have not been published elsewhere. The authors well described the methodology and results with sufficient details. This study has received ethical clearance from the Ethics Review Committee of Fujian Province Maternity and Child Health Hospital (approval no. 073, 2019), and this study has been conducted with adherence to the Declaration of Helsinki. This manuscript has provided new information on β-thal disease by highlighting the role of the hsa-miR-19b-5p in ameliorating the β-thal through its direct action on BCL11A to downregulate BCL11A and thus increase the HbF level in β-thal patients. The potential of hsa-miR-190b-5p as a biomarker for β-thal was evaluated by the authors, who suggested that hsa-miR-190b-5p is similarly important as HbA2 for discriminating pediatric β-thal patients from healthy controls. The authors also proved the capability of hsa-miR-19b-5p direct binding to BCL11A. The authors have clearly and thoroughly discussed their findings with relevant references. Some grammatical errors may be corrected prior to publication. 

1) Page 8, recheck this sentence: "Compared with healthy controls, the levels of red blood cells (RBC) and hemoglobin (Hb) in pediatric β-thal patients (p<0.001)."; 

Response: We are very appreciated for this comment. We have rechecked this sentence “Compared with healthy controls, the levels of red blood cells (RBC) and hemoglobin (Hb) in pediatric β-thal patients (p<0.001).” and change it to “Compared with healthy controls, the levels of red blood cells (RBC) and hemoglobin (Hb) were significantly lower in pediatric β-thal patients (p<0.001)” in Page 8, line 16.

2) Page 12, recheck this sentence: "The above data hinted that that decreased hsa-miR-190b-5p has potential as a diagnostic biomarker for pediatric β-thal."; 

Response: We are very appreciated for this comment. We have rechecked this sentence “The above data hinted that that decreased hsa-miR-190b-5p has potential as a diagnostic biomarker for pediatric β-thal.” and change it to “The above data hinted that decreased hsa-miR-190b-5p could be used as a potential diagnostic biomarker for pediatric β-thal.” in Page 15, line 13.

3) Figure 1, recheck this sentence: "Regulation of hsa-miR-190b-5p on the garget gene BCL11A.".

Response: We are very appreciated for this comment. We have rechecked this sentence “Regulation of hsa-miR-190b-5p on the garget gene BCL11A.” and change it to “Regulation of hsa-miR-190b-5p on the target gene BCL11A.” in revised Figure 1.

---

## [Decision Letter · Decision Letter 1]

30 Aug 2023

PONE-D-23-10873R1The clinical value of hsa-miR-190b-5p in peripheral blood of pediatric β-thalassemia and its regulation on BCL11A expressionPLOS ONE

Dear Dr. Huang,

Thank you for submitting your manuscript to PLOS ONE. After careful consideration, we feel that it has merit but does not fully meet PLOS ONE’s publication criteria as it currently stands. Therefore, we invite you to submit a revised version of the manuscript that addresses the points raised during the review process.

We look forward to receiving your revised manuscript.

Kind regards,

J Francis Borgio, Ph.D.,

Academic Editor

PLOS ONE

Journal Requirements:

Reviewers' comments:

Reviewer's Responses to Questions

**Comments to the Author**

1. If the authors have adequately addressed your comments raised in a previous round of review and you feel that this manuscript is now acceptable for publication, you may indicate that here to bypass the “Comments to the Author” section, enter your conflict of interest statement in the “Confidential to Editor” section, and submit your "Accept" recommendation.

Reviewer #2: All comments have been addressed

Reviewer #3: All comments have been addressed

2. Is the manuscript technically sound, and do the data support the conclusions?

Reviewer #2: (No Response)

Reviewer #3: Yes

3. Has the statistical analysis been performed appropriately and rigorously? 

Reviewer #2: (No Response)

Reviewer #3: Yes

4. Have the authors made all data underlying the findings in their manuscript fully available?

Reviewer #2: (No Response)

Reviewer #3: Yes

5. Is the manuscript presented in an intelligible fashion and written in standard English?

Reviewer #2: (No Response)

Reviewer #3: Yes

6. Review Comments to the Author

Reviewer #2: (No Response)

Reviewer #3: Nice paper

would have been interresting to see if miR190 was not influenced by anemia out or thalassemia

Minor comments :

Page 3 line 24 “…In recently, several…” what does this means?

Page 13 line 3“…was negative regulated…” do you mean negatively regulated?

7. PLOS authors have the option to publish the peer review history of their article (what does this mean?). If published, this will include your full peer review and any attached files.

Reviewer #2: No

Reviewer #3: **Yes: **Claude LAMBERT

---

## [Author Response · Author response to Decision Letter 1]

1 Sep 2023

Response letter

[Sept 1 2023]

Dear Editor and reviewers,

First of all, thank you very much for giving us an opportunity to revise our manuscript. We appreciate editor and reviewers for the comments concerning our manuscript entitled “The clinical value of hsa-miR-190b-5p in peripheral blood of pediatric β-thalassemia and its regulation on BCL11A expression” (ID: PONE-D-23-10873R1). Those comments are all valuable and very helpful for revising and improving our paper. We have studied comments carefully and have made correction which we hope meet with approval. We have thoroughly revised the manuscript accordingly, and changes corresponding to the issues were highlighted in Red in the revised version.

 We very much appreciated the valuable comments and suggestions made by the reviewers. We hope that the changes have adequately addressed their concerns. Thank you very much for the time and effort you have spent to help us improve this manuscript.

Sincerely yours,

Hailong Huang

Medical Genetic Diagnosis and Therapy Center of Fujian Maternity and Child Health Hospital College of Clinical Medicine for Obstetrics & Gynecology and Pediatrics, Fujian Medical University, Fujian Provincial Key Laboratory of Prenatal Diagnosis and Birth Defect, Fuzhou 350001, Fujian Province, China

Email: huanghailong@fjmu.edu.cn

Journal Requirements:

Response: All references cited in our manuscript have not been retracted. All references cited in our manuscript have been formatted according to the requirements of PLOS ONE.

Responses to Reviewer #3’s comments

Reviewer #3: Nice paper

would have been interesting to see if miR190 was not influenced by anemia out or thalassemia

Response: We are very appreciated for this comment. In the future, we will be collaborating with other research groups to investigate the expression and clinical value of hsa-miR-190b-5p in other hematological diseases, such as leukemia, multiple myeloma, aplastic anemia.

Minor comments:

Page 3 line 24 “…In recently, several…” what does this means?

Response: The meaning of this sentence is that BCL11A is key regulator for HbF expression, and other transcription factor (KLF1, KLF1 and MYB) could impact HbF expression via regulation of BCL11A. This sentence is a bit vague, we've corrected it.

Page 13 line 3“…was negative regulated…” do you mean negatively regulated?

Response: Thanks for the reminder. It was a grammatical error and we've corrected it.

---

## [Decision Letter · Decision Letter 2]

11 Sep 2023

The clinical value of hsa-miR-190b-5p in peripheral blood of pediatric β-thalassemia and its regulation on BCL11A expression

PONE-D-23-10873R2

Dear Dr. Huang,

We’re pleased to inform you that your manuscript has been judged scientifically suitable for publication and will be formally accepted for publication once it meets all outstanding technical requirements.

Kind regards,

J Francis Borgio, Ph.D.,

Academic Editor

PLOS ONE

Additional Editor Comments (optional):

The revised MS can be accepted

Reviewers' comments:

Reviewer's Responses to Questions

**Comments to the Author**

1. If the authors have adequately addressed your comments raised in a previous round of review and you feel that this manuscript is now acceptable for publication, you may indicate that here to bypass the “Comments to the Author” section, enter your conflict of interest statement in the “Confidential to Editor” section, and submit your "Accept" recommendation.

Reviewer #3: All comments have been addressed

2. Is the manuscript technically sound, and do the data support the conclusions?

Reviewer #3: Yes

3. Has the statistical analysis been performed appropriately and rigorously? 

Reviewer #3: Yes

4. Have the authors made all data underlying the findings in their manuscript fully available?

Reviewer #3: Yes

5. Is the manuscript presented in an intelligible fashion and written in standard English?

Reviewer #3: Yes

6. Review Comments to the Author

Reviewer #3: Nice paper

Substantially improved by following reviewers comments

ok for publication in PLOS 1

The paper is now ready to be published

7. PLOS authors have the option to publish the peer review history of their article (what does this mean?). If published, this will include your full peer review and any attached files.

Reviewer #3: **Yes: **Claude LAMBERT
